# The importance of organisational culture for health system resilience: A qualitative analysis of factors that supported health care workers in Germany during the COVID-19 pandemic

Heide Weishaar[1]*, Megan Evans[1], Souaad Chemali[2], Eloisa Montt Maray[1],
Rike Böttcher[1], René Umlauf[1], Susan Abunijela[3], Nadine Muller[1],
Barbara Buchberger[4], Brogan Geurts[1], Hanna-Tina Fischer[1], Charbel El Bcheraoui[1]

1 Evidence-based Public Health, Centre for International Health Protection, Robert Koch Institute, Berlin, Germany, 2 Preparedness and Operations Support Unit, Centre for International Health Protection, Robert Koch Institute, Berlin, Germany, 3 Infectious Respiratory Disease, Abteilung 3, Department of Infectious Disease Epidemiology, Robert Koch Institute, Berlin, Germany, 4 Robert Koch Institute, Berlin, Germany

* WeishaarH@rki.de

**Data Availability Statement:** The data underlying the results of this study cannot be publicly shared

## Abstract

Health system resilience is defined as the ability of a system to prepare, manage, and learn from shocks. This study investigates the resilience of the German health system by analysing the system-related factors that supported health care workers, a key building block of the system, during the COVID-19 pandemic. We thematically analysed data from 18 semi-structured interviews with key informants from management, policy and academia, 17 in-depth interviews with health care workers, and 10 focus group discussions with health care workers. We identified six categories of factors which supported health care workers during the pandemic: (1) appreciation and encouragement; (2) team support and communication; (3) collaboration across professions and institutions; (4) informational, material and technical support; (5) leadership and participation; and (6) creativity and innovation. The analysis highlights the importance of systemic and organisational factors for maintaining health system resilience. The need to improvise and develop pragmatic solutions to deal with COVID-related challenges offered ample opportunities for participation, self-determination, innovation, and alternative work processes. Learning from adaptations made during the pandemic and implementing sustainable organisational and systemic changes to increase participation, conducive organisational cultures, and non-hierarchical working styles hold promising potential for health system transformation and health system strengthening.

## Introduction

Health system resilience is defined as a health system's ability to "anticipate, prevent, prepare for, absorb and adapt in response to, and recover from a wide variety of shocks and stressors while delivering high-quality individual and population health services as needed" [1].

due to confidentiality agreements with participants and the data protection regulations of the Robert Koch Institute (RKI). Data will be made available upon reasonable request, subject to compliance with the RKI's confidentiality and data protection policies. Any shared data will be pseudonymized and anonymized to protect the privacy of participants, in line with informed consent agreements. For data access requests or inquiries, please contact Datenschutz@rki.de.

**Funding:** CEB received funding by the German Federal Ministry of Health through the Global Health Protection Programme (Kapitel 1505 Titel 68601).The funders had no role in study design, data collection and analysis, decision to publish, or preparation of the manuscript. Information about the German Federal Ministry of Health can be found here: https://www.bundesgesundheitsministerium.de/en/index.html.

**Competing interests:** The authors have declared that no competing interests exist.

Resilience is a complex construct and can be applied to different contexts, including to individuals, groups, organisations, and systems [2]. While political sciences, environmental sciences, history, disaster planning and other fields have largely described resilience as a system's ability to absorb disturbances and perform when faced with shocks [3], psychology primarily uses the term to describe an individual's ability to deal with, and recover from, a shock or strain [4, 5]. It can thus be argued that resilience can be assessed at both individual and system level. Blanchet et al.'s resilience framework categorises the ability of a health system to manage a shock according to three capacities [6]: (i) the absorptive capacity ensures that the system can maintain health services using previously existing resources and capacities; (ii) the adaptive capacity refers to the system's ability to maintain health services with fewer or different resources by making organisational adaptations; and (iii) the transformative capacity describes the system's ability to transform its functions and structure to respond to changes [6]. While recognising the need for a health system to absorb, adapt, and transform, recent debates on health system resilience also increasingly highlight the need for a health system to further evolve through learning from shocks [7]. It is anticipated that such evolution can eventually lead to increases in the system's resilience and an improved capacity to deal with future threats of a similar nature [7]. This literature recognises that a shock requires a system to substantially change and that it is therefore unlikely that the system will simply "bounce back" to its original state. Instead, a resilient health system will transform by implementing lasting systemic change and improvements after the shock [7].

Studying resilience can be of practical value as it provides insights into the ways in which components of a system can facilitate the successful management of a shock for the system as a whole and the implementation of system-wide adaptations. If what contributes to resilience is understood, it is possible to support a system in advance of, and during, a shock [8] as well as in post-shock transformation. Studies that investigate how a health system behaves during a shock can thus help to identify potential for strengthening the system. Given that the COVID-19 pandemic resulted in an–in modern times–unprecedented shock to health systems worldwide, it provides a unique opportunity to study health system resilience. We propose here to study health system resilience by focusing on the ways in which changes in one of the health system building blocks, namely the health workforce, have system-wide effects.

Our study investigates health system resilience by analysing the system-related factors that supported health care workers (HCWs) during the COVID-19 pandemic [9]. The health workforce as one building block of a health system [10] lends itself as an object of study of resilience in pandemic times for two significant reasons. First, the health workforce plays a critical role in ensuring health services are maintained during pandemics, which has once again become apparent during COVID-19 [9]. Resilient HCWs are thus key contributors to health system resilience [9]. Second, empirical analyses of the situation of HCWs during the COVID-19 pandemic provide evidence that HCWs were greatly affected with impact on their psychological wellbeing [11–15]. This means that the pandemic posed a challenge to HCWs' resilience. Germany provides a good case to study health system resilience with a specific focus on the health workforce as a building block as the German health system had to deal with a comparatively high number of COVID-19 cases, while also struggling with a chronic shortage in HCWs [16–18]. Box 1 presents a short summary about the COVID-19 pandemic, including the number of cases and fatalities among the general population and among HCWs, impact on the health system. Understanding which factors of the German health system supported HCWs during the pandemic can retrieve lessons that will help prepare the system and those working within it for future public health crises.

Box 1. COVID-19 pandemic in Germany

As of 26 August 2024, since the start of the pandemic 38,875,986 individuals in Germany were infected with SARS-CoV-2, and 183,752 died of COVID-19. Out of these, Germany reported a total of 16,012,827 COVID-19 cases and 32,074 deaths in the period during which the data for this study was collected (April–December 2022) [19]. An analysis of SARS-CoV-2 infections according to professional groups shows that people working in the health sector reported a higher prevalence of infection than the average of all respondents across all professional groups (14.1% vs 10.5). According to this analysis, employees in the healthcare sector have a 1.68-fold higher risk of SARS-CoV-2 infection compared to workers in other occupations after adjusting for other possible influencing factors (e.g. age, gender, education and vaccination) [20]. A study analysing data from the institution for statutory accident insurance and prevention in the health and welfare services indicates that 376,557 employees from the sector reported COVID-19 as an occupational illness until the end of 2022, with nursing staff reporting the highest numbers. 4,730 occupational illness cases were hospitalised, and 169 died [21].

## Materials and methods

We conducted focus group discussions and individual interviews with HCWs as well as individual interviews with key informants involved in decision-making around health service provision as part of an international, mixed-method study assessing HCWs' experiences during the COVID-19 pandemic. Qualitative methods were chosen as they allowed a detailed investigation of the views of HCWs themselves as well as of key informants familiar with the situation of HCWs and in-depth insights into the system-related factors that supported HCWs during the COVID-19 pandemic [22]. Interviews were used to elicit detailed descriptions of experiences and views from both HCWs and key informants and helped to gain a variety of perspectives. During the FDGs, some aspects that had been raised or had emerged as topics of contention during the interviews were discussed. The FDGs thus provided more detail on selected aspects, helped to clarify some issues, and shed light on controversies and disagreement. HCWs were recruited between 1 February and 30 November 2022 through a quantitative survey on stress and coping among HCWs during the COVID-19 pandemic which was conducted as part of the overarching study. Survey participants were recruited via the health facilities in which they worked or via professional networks and organisations. HCWs who completed the survey and expressed an interest to participate in the qualitative arm of the study were contacted individually via email. HCWs were included in the study if they were: (i) aged 18 or above, and (ii) had been working in a health facility during the COVID-19 pandemic.

Data were collected between April and December 2022. HCWs who participated in the interviews and FGDs included medical doctors, nurses, allied health workers, nursing students, and other staff working in health facilities during the pandemic. We invited a wide range of HCWs working across all levels of health care (primary, secondary, tertiary) and in a range of health facilities in Germany, including private practices, hospitals, and care homes. Potential participants received information about the study and were given the option to either participate in an individual interview or a FGD.

Key informants included political decision-makers, health facility managers, representatives of professional associations, representatives of health care providers, academics, and other

individuals with decision-making functions. Key informants were sampled through an online search where potential interviewees were identified from media and other reports. Consecutively, individuals who had commented on the situation of HCWs during the pandemic were listed as potential participants. Individuals were considered as key informants if they were (i) aged 18 years or above, and (iii) involved in decision-making around health service provision. Purposive sampling was then undertaken to ensure a diverse selection of key informants from various regions, areas of work, genders, and roles. Selected individuals were contacted between 1 April 2022 and 30 November 2022 via email, provided with comprehensive study information and asked to participate in a semi-structured interview.

All participants were required to be 18 years old or above and to provide voluntary informed written consent. Information sheets and informed consent forms were shared with the participants before the interview or FGD. Other than interactions about logistical arrangements, no relationship was established prior to the interview of FGD. Interviews with key informants were conducted online via videoconference or face-to-face, while all HCWs' interviews and FGD were conducted online. HCWs received a 25 Euro voucher as compensation for participation in interviews or FGDs, while key informants did not receive compensation. Interviews and FGDs were jointly conducted, usually with one senior researcher acting as interviewee/facilitator (HW, RB, BB) and a junior researcher or administrative assistant as a note taker (RB, SA). All interviewees and facilitators were members of the study team and employed as staff at the institution leading the study. Interviews with HCWs and key informants took on average 55 minutes (min: 37 minutes, max: 71 minutes), whereas FGDs lasted on average 70 minutes (min: 57 minutes, max: 79 minutes). All interviews and FGD were conducted in German and audio-recorded. Field notes were taken for each interview and FGD. Verbatim transcription and translation into English was performed by an external company to facilitate analysis among the international study and the team of researchers. Data collection was concluded as data saturation was reached.

Topic guides were developed based on a literature review [11] and covered the following topics: personal experiences, challenges, coping strategies, factors supporting or hindering HCWs in coping with challenges, support needs, and recommendations for support. Guiding questions were defined for each section and tailored to the data collection method and target group (FGD with HCWs, interviews with HCWs, and interviews with key informants). Topic guides were amended as data collection progressed to account for emerging themes from previous interviews or FGDs and clarify conflicting data.

All data were analysed according to thematic content analysis [23] using NVIVO software 1.7. As a first step, the coding team members familiarized themselves with the transcripts, reading and rereading them, and taking margin notes. A codebook was then developed using a priori, or pre-defined codes that came from the interview topic guides, and in vivo codes that were identified from wording used by respondents [24]. The development of the codebook was an iterative, collaborative process. As part of this, a random sample of five interviews was independently coded by three researchers. Coding was discussed, and the codebook was revised thoroughly according to consensus. The final codebook included a definition for each code, specifications for when to use it, and specifications for when not to use it. All transcripts were systematically coded by six members of the study team (ME, EMM, RB, RU, BB, HW), applying the revised codebook. One third of all transcripts was double coded to ensure coding was consistent. Consistency checks were performed by a senior researcher (HW). The analysis involved the integration of similar thematic coding, with findings being subsequently triangulated across different data sources for each theme. As part of the analysis, particular attention was paid to identifying evidence that appeared to contradict the explanations being developed and to considering alternative explanations. Over the course of data collection and analysis, all

members of the research team reflected on their own positioning with regard to the research topic. We particularly reflected on the potential impact of the following aspects on data collection and analysis: (i) our professional backgrounds as researchers and (for selected members of the research team) as health professionals; (ii) our affiliation with an institution that communicated extensively during the COVID-19 pandemic and provided evidence which informed political decisions with major impact on HCWs; (iii) the participants' potential intentions to promote a personal or political agenda or to, retrospectively, recount situations in a way that supported their views and opinions.

## Results

A total of 60 HCWs participated in the study, with 43 HCWs taking part in 10 FGDs and 17 HCWs in individual interviews (Table 1). Additionally, individual interviews were conducted with 18 key informants. Throughout data collection, it became apparent that HCWs struggled with multiple challenges during the COVID-19 pandemic. Listening to their accounts of what supported them to successfully tackle these challenges and maintain resilience, five overarching, interconnected themes could be identified. The five themes related to the work context, the culture of the organisations HCWs worked in and the health system. In addition, individual factors were identified as influencing how HCWs managed COVID-related challenges. We therefore give a brief summary of the individual factors that were mentioned as mediating HCWs' reactions prior to providing a detailed description of the five work- and system-related themes. Participants' quotes are provided as illustrative examples to highlight specific aspects of each theme. Depending on the preferences of the respective interviewee or FGD participant, each quote is allocated to the participant's personal data, or only generic information is given that the quote was provided by a study participant.

**Table 1. Socio-demographic data of HCWs who participated in the study.**

|  |  | FGD (n) | Interview (n) | Total (n) |
|---|---|---|---|---|
| Gender | Male | 18 | 5 | 23 |
|  | Female | 25 | 12 | 37 |
| Age | 20–29 | 3 | 2 | 5 |
|  | 30–39 | 11 | 2 | 13 |
|  | 40–49 | 7 | 3 | 10 |
|  | 50–59 | 17 | 8 | 25 |
|  | 60+ | 5 | 2 | 7 |
| Levels of health care | Primary | 6 | 7 | 13 |
|  | Secondary | 9 | 5 | 14 |
|  | Tertiary | 28 | 5 | 33 |
| Profession | Therapist | 14 | 5 | 19 |
|  | Nurse | 6 | 8 | 14 |
|  | Doctor | 10 | 1 | 11 |
|  | Manager | 5 | 1 | 6 |
|  | Medical assistant | 3 |  | 3 |
|  | Administrator | 1 | 1 | 2 |
|  | Pharmacist | 1 |  | 1 |
|  | Social worker | 1 |  | 1 |
|  | Paramedic |  | 1 | 1 |
|  | not answered | 2 |  | 2 |

HCWs reported a number of individual factors and strategies that were of importance in dealing with the pandemic situation. HCWs recalled an inner strength that carried them through the pandemic. One respondent referred to being "resilient" (paramedic, male, 28 years, tertiary level), another described an "inner conviction that one can do one's part" (medical assistant, female, 35 years, primary level) as a supportive trait. A proactive attitude was further mentioned as conducive to successfully handle difficulties experienced. Believing in oneself and a sense of self-efficacy were further highlighted to ease the situation. A focus group discussant expressed:

> "What helped us get through this time? I believe it was the faith in each other. [. . .] faith in people and their work. Having hope that things will improve and we will eventually get through this time." (focus group discussant)

Respondents further stressed the importance of recreation. Walking in the outdoors, being in nature, doing sports, and other recreational activities were mentioned as helpful strategies to maintain resilience. Family members and friends as well as spending time with others in ways that were in line with pandemic restrictions (e.g. online, with the family or as a household) emerged as important sources of support and comfort. Distancing oneself from work and the pandemic events, and prioritising time for social interactions and recovery were mentioned as useful mechanisms to deal with the challenges and demanding situations at work. HCWs reported that keeping boundaries between work and private life was often challenging given the high work demands, the absence of colleagues and existing staff shortages. Deciding to do so was described as an act of self-care, as the following quote by a focus group discussant illustrates:

> "For me, it was, well, self-protection to say at some point: 'Stop, I'm now keeping my regular working hours, I'm no longer working until I drop, but have now set myself fixed working hours, which I also try to keep as a rule.' Simply as self-protection, otherwise you really collapse." (physiotherapist, female, 56 years, primary level)

## Appreciation and encouragement from employers and patients

HCWs mentioned appreciation and encouragement, e.g. from patients and their relatives, line and facility managers and others, as a key factor that supported them to continue providing health care despite the COVID-related challenges and adversities. Thankfulness, positive interactions and small gifts of appreciation were perceived as great boosters. Furthermore, positive medical outcomes, e.g. patients being released from intensive care units, being discharged or successfully treated also greatly motivated respondents to keep going. Despite having done their job for years, sometimes decades, HCWs directly involved in medical, therapeutic or nursing tasks reported to be moved and motivated by interpersonal interactions with patients, as the following quote illustrates:

> "I can remember elderly ladies whom I visited, COPD [with chronic obstructive pulmonary disease], with oxygen, who still bravely wore their masks when I came over. Those were experiences where I thought: 'Yes, we can do this.' That was definitely positive." (physician, male, 33 years, primary level)

Similarly, HCWs highlighted instances where they had received appreciation and empathy for managing challenging situation. Small signs of appreciation seemed to go a long way, with

individual respondents remarking that their direct line managers had left handwritten notes or given them personalised vouchers. It seemed that rather than the gifts themselves, what was most appreciated was the effort, consideration, empathy, and care that was demonstrated through the gift-giving:

*"I have been working in this hospital for 30 years and I have never seen such a great manager. She really takes care of the staff. For example, everyone received a handwritten Christmas card, which is something that I have never seen before. And we all were given key chains of different colours. These are trivialities, but it shows that she really thinks of the staff. She takes care of the staff, even when she is under pressure herself. You can tell because she not just manages staffing to fill in gaps, but she also tries to make us stick together as a team. And I think she does a good job." (interviewee)*

### Team support and communication

Respondents who directly worked with patients in a medical, therapeutic or nursing capacity stressed that it was important to maintain communication within the team and find ways to connect face to face despite the social distancing measures. Message platforms like WhatsApp were increasingly used to exchange specific information about patients and tasks, but also to ensure regular team interactions. Respondents repeatedly mentioned that not being able to meet colleagues in person complicated work flows, and also made it difficult to get a feel for how colleagues were doing. Despite hygiene measures, some teams therefore decided to re-establish face-to-face meetings.

*"So, we moved into a giant room, all windows open, with winter coats, doesn't matter. Because it was clear that we had to see each other. Otherwise, I couldn't sense how people were doing. And you need to. You need to see: Do you need a break, should we send someone else, shall we do it on another day?" (physiotherapist, female, 60 years, tertiary level)*

Respondents repeatedly highlighted that the positive interactions that they had with colleagues and the mutual support that they provided within collegial teams were of immense importance. This seemed to particularly be the case for HCWs who directly interacted with patients, and even more so those who encountered stressful situations at work. Many respondents said that the teams had become stronger as colleagues often supported each other well and were usually the first point of call when having to develop solutions to new challenges. Often, the support provided was of a practical nature, with colleagues taking over shifts or tasks. Moreover, team interactions also seemed to provide support with regard to dealing with unexpected situations and uncertainty. An interviewee recalled how they and their colleagues had helped each other out with regard to developing strategies to handle patients who were reluctant to wear masks:

*"But we also remained extraordinarily strong in the team. We sought arguments among ourselves about how to deal with people who come in and say: 'No, that's nonsense for me.'" (physiotherapist, male, 60 years, primary level)*

As HCWs increasingly faced emotionally challenging and distressing situations, they seemed to rely on collegial support, with some reporting the establishment of new formats for peer support (e.g. debriefing sessions) in order to enable interactions about pandemic-specific challenges. Responses suggest that, due to providing a space and time for exchange about

mutual experiences and peer support, the interactions with colleagues (as opposed to interactions with external individuals, including family members, friends or professional supporters like coaches or therapists) were able to provide a very specific, valuable kind of support. Many respondents highlighted that being able to communicate with others who had similar experiences and sharing feelings within the team was of particular value:

*"You have to say that with the teams it's often the case that a lot of things you don't want to discuss outside the team, but rather within the team. That's why I started out, for example, with the habit that after a resuscitation, after every resuscitation or after a death that was fulminant in a young patient, we all sit down together and have coffee. And everybody speaks again and expresses their feelings if he or she wants to. That was also something new, that didn't exist before." (senior doctor teaching hospital)*

Chats with colleagues and as a team about what had happened seemed to allow HCWs to discuss cases, review the lessons learned, and *"to free oneself a bit from one's own mistakes" (nurse, female, 30 years, secondary level).*

## Collaboration across professions and institutions

Another factor that was highlighted as crucial to improving the work situation was the collaboration across professions and institutions which in many cases was reported to intensify during the COVID-19 pandemic. Particularly older respondents and those in senior positions positively recalled that staff of various professional backgrounds worked together in a constructive, respectful and efficient manner. One HCW enthusiastically described how they had collaborated with other HCWs from a range of disciplines in order to develop solutions to COVID-related challenges:

*"For one thing, what was really outstanding in these two years was the interdisciplinary collaboration. Well, as I said, I had almost 100 different employees! [. . .] Well, all this cooperation within the hospital was incredible, I have to say. Even all these service departments that you need to provide material, whether it's pharmacy or also in-house transport, that was a cooperation I had never experienced before. It was so intense and it all kind of worked itself out." (interviewee)*

Respondents highlighted how increased collaboration had helped members of each discipline to better understand their colleagues' tasks. In one general practice, the mutual learning was facilitated by team role plays, which allowed team members to experience what their colleagues were going through:

*"We also do regular roleplays, where we put ourselves into each other's shoes. So, the medical assistants will re-enact how patients behave at receptions, and us doctors will enact for each other how patients behave in the examination room, so that we have a little more understanding for each other." (HCW, male, 55 years, primary care)*

Closer collaboration across disciplines and institutions seemed to have positive public health and medical effects. Communication and collaboration between different professional groups contributed to the containment of the virus, helped to clarify specific therapeutic and medical questions and tackle challenges that emerged as a result of the pandemic, e.g. regarding the treatment of severe COVID-19 cases and chronically ill patients, the provision of personal protective equipment (PPE) or the implementation of isolation guidelines. As a key

informant described, health care providers who were previously competing for patients and services were reported to work together to overcome the challenges posed by the pandemic:

> *"This means that the hospitals that were geographically connected in a certain region really worked closely together. They agreed on who would possibly take on which tasks in order to cope as well as possible with the available capacities. In many cases we found that competitors actually became cooperation partners. [. . .] One reason was the common challenge, where we said that there is nothing to negotiate here. We have to make all our capabilities available for the protection of the population. So everyone was clear about what the priority here is now."* (chairman of a board)

Respondents described that the realisation that all shared the same concerns and faced similar challenges made them feel *"a bit like a big family"* (health facility manager, male, 58 years, primary level) or *"like going to war together"* (nurse, female, 58 years, tertiary level) and led to collaboration across professions and institutions. This cessation of competition, the descriptions of a *"team spirit"* (president of the medical association of Lower Saxony) and *"sense of a community"* (physician, male, 28 years, tertiary level) pertained several interviews and FGDs. Respondents often contrasted the increased level of intra- and interorganisational collaboration with the hierarchical, siloed way of working in health care that had been the norm pre-COVID, when the different professions, sectors and institutions had largely been working separately from each other with little interaction. Respondents further highlighted that the improved collaboration continued, suggesting that the pandemic had sparked some more sustainable changes in working across disciplines and institutions.

## Informational, material, and technical support

All HCWs, independent of level of health care or profession, highlighted the importance of informational, material, and technical support. While many noted that such support was often not provided, individual respondents recalled instances where they benefited from receiving information, provision of materials, and technical training. Regular, clear, and evidence-based updates about the pandemic situation, the transmission of the SARS-COV-2 virus, and the effective measures to prevent transmission seemed to be more frequently received by HCWs in large health facilities of the inpatient sector (as opposed to the ambulatory sector and private practices). These updates helped HCWs to understand the situation and the risks, enabling them to fulfil their roles as information brokers when patients or relatives had questions. Not having access to reliable information, on the other hand, caused stress and insecurity:

> *"Well, for me it was more a matter of somehow trying to get hold of information, because it was simply stupid for the patient and for myself to stand there in the morning and not being able to provide real information to the patient or the staff, also for myself."* (focus group discussant)

Accurate, up-to-date information about policies, guidelines, and hygiene regulations was also perceived as useful as it reassured HCWs that they were in line with the regulations. This was of particular importance as regulations were continuously and rapidly amended. What seemed even more important than information on governmental regulations were transparent communication and clear instructions from the management of the health facility, so that HCWs understood what was expected of them and had a clear direction.

In terms of material and technical resources, it was positively remarked when employers had provided sufficient PPE, so that HCWs could effectively protect themselves from infection

and felt safe. Also, specific expertise and experience in crisis management were perceived as immensely useful. A small number of HCWs reported that their employer had ensured that they were trained in crisis management, COVID-19 testing and hygiene procedures. Such technical support was perceived as reassuring, with one HCW reporting that it had given her *"great certainty when it comes to treating COVID patients"* (nurse, female, 54 years, primary level). A focus group respondent recalled emergency simulation exercises that had been conducted in their health facility to prepare the staff for health crises:

> *"We started relatively early to build up a COVID setting [as part of the simulation exercises]. That means, what do we do if we have to treat a COVID-positive patient, what are the processes? We acted out how we would dress each other, how to put masks and visors on, et cetera. And this preparation was a great help for us."* (consultant in anaesthesia, male, 57 years, tertiary level)

Selected larger health facilities supported HCWs via other pragmatic steps, including the provision of rest areas, the delivery of food when HCWs had to work long shifts or had only short breaks between shifts, and impromptu solutions with regard to child care. A small number of HCWs reported that they had been offered psychological support, yet this did not seem to be the standard support provided.

## Leadership and participation

Respondents stressed that the way in which their employers and health facility management handled the pandemic situation and supported employees was a crucial determinant of their ability to handle work-related challenges. In a number of instances, respondents reported about changes in organisational culture and increased participation. While a supportive leadership style and participatory decision-making were identified as key supporting factors, the analysis also revealed that good leadership, employer's support, and participation were frequently lacking, adding to the burden that HCWs had to deal with during the pandemic.

A key factor that was positively noted were opportunities to interact with the employer and to participate in decision-making. HCWs stressed that being informed and kept up-to-date by the management of the health facility they worked in helped them to handle challenging work situations. Similarly, those in leadership positions emphasised the importance of good communication, of *"stay[ing] close to our staff"* (manager facility) and of *"getting feedback from employees"* about what they needed and how they could be supported (manager health facility, female, 59 years, primary level). Some respondents claimed that participatory decision-making had increased in their health facility during the pandemic, and that decisions had been taken under consideration of the factors that were relevant to those affected. A key informant recalled:

> *"We always made these decisions as a team. Never alone, always after considering many factors. Namely, social factors, infection control, and in the end, we made our decision on these grounds."* (manager health facility)

Some respondents suggested that the crisis had facilitated shifts in organisational culture, as health facility staff were allowed to break from previously existing hierarchies, divisions, and traditional routines and act more flexibly. Recalling how hygiene specialists had started to provide guidance to the rest of the hospital and had taken over from those usually in charge, a focus group discussant highlighted how this *"softened a good part of these previously quite rigid hierarchies"* (nurse, male, 57 years, tertiary level). Similarly, other respondents spoke about the

breaking down of *"principalities within the hospital"* (senior doctor, teaching hospital) and pre-viously existing *"silos"* (manager of nursing, hospital). A small number of respondents posi-tively remarked the flexibility of employers and facility management. One doctor voiced surprise about his facility management's ability to break up *"absolutely old, entangled structures within a very short time in order to be able to react to this situation. [. . .] I would never have thought that it was possible [. . .] Well, the decisions of the hospital operations management were, for example, which department to close, which operating rooms to cancel immediately so that [. . .] staff had to be shifted here and there. [. . .] That was for many heads of department an utter absurdity that something like that was even proposed. And here it was decided very quickly, so that everything can continue. [. . .] Well, really quite pragmatic, tangible things."* (senior doctor, teaching hospital).

The willingness to find pragmatic solutions and a more participatory decision-making style seemed to be particularly appreciated with regard to finding solutions to ethically challenging situations. A focus group discussant who worked in a care home reported how their team had tried to find an ethical, feasible solution to handle residents with dementia who were tested positive for COVID-19:

> *"Hierarchical thinking no longer played a role, or very little. And basically, if you wanted to live democracy or really wanted to experience it, you could actually do that quite well during the pandemic, because we had important questions that you had to evaluate. For example, what we were very concerned about at the very beginning of the pandemic was how to deal with people with dementia who are infected. [. . .] And so, we basically had an ethical discus-sion about how to deal with it. [. . .] Of course, you don't make a decision like that on your own. That was quite helpful in the facility, that we found common values, where we could also deal ethically and morally well [. . .]. We had to clarify it by ourselves." (health facility man-ager, male, 58 years, primary level)*

Considerations which weighed the regulatory situation against ethical and pragmatic fac-tors in some instances resulted in HCWs and management of both ambulatory and inpatient facilities violating governmental rules and policies. Decisions such as not wearing full PPE when dealing with patients with dementia or in order to provide the necessary treatment or allowing relatives to say goodbye to their loved ones prior to dying, were described as acts of civil disobedience. Such resistance seemed to result from assessments made in collaboration with the team that regulations could only be implemented with major negative consequences for staff or patients and that the benefits of pragmatic alternatives that were implemented based on participatory decision-making by far outweighed the risks.

## Creativity and innovation

Respondents across levels of care and professions mentioned many instances when they inno-vated and developed creative solutions to COVID-related challenges, ranging from sowing facemasks from bedsheets, to producing self-made disinfectant, initiating projects to raise funds, building make-shift divisions with plexiglass panes or establishing digital solutions for communication in order to maintain social distancing while allowing interaction between patients and visitors, and developing new methods of documentation and collaboration. Such innovations were almost exclusively developed through bottom-up approaches, with HCWs taking initiative based on their perceptions of need and independent of institutional or govern-mental support. One innovative example of collaborative patient care was provided by one general practitioner (GP) who reported that they and their colleagues had agreed between

themselves that only one local GP would visit all quarantined COVID-positive patients, reducing the other GPs' risk of infection and thus the risk of health care services breaking down. The need to improvise seemed to release creativity, lateral thinking, and innovation which in previous work situations had not been needed or requested. Respondents working both in the ambulatory and the inpatient sector stressed that innovative solutions were often implemented very quickly, and that departments and colleagues were willing to adapt suggestions and learn from each other. Faced with a common challenge and the shared goal of coping with the pandemic situation while maintaining quality care, HCWs felt liberated to deviate from established routines and develop alternative ways of doing things:

> "Yes, the breaking away of routine is obvious in such a situation, in such a crisis situation. Something completely new rolls in on you. But [. . .] this interaction of the diverse employees, that made it easy. They were really motivated and [. . .] we had a common goal. [. . .] And we had to, we had to structure ourselves within a very short period of time to take good care of the patients and to also—we had to take care of ourselves as well. And I think it was this motivation and also the unknown, this challenge, that made it possible for us to quickly find workflows." (interviewee)

Respondents highlighted that they "changed processes, created new teams, changed responsibilities and [. . .] improvised" (manager of nursing, hospital) claiming that "hospitals are masters of improvisation" (senior official in Ministry of Health in Lower Saxony).

In some instances, creative solutions were found that involved not only HCWs and facility management, but also individuals and organisations outside the facility. One HCW described how mobilising relatives to support the care of care home residents had resulted in a win-win situation:

> "We did this in [. . .] those areas where we really didn't have any staff left. So we asked those relatives, where we knew that they've always been quite involved and also interested in nursing care, whether they could support us by taking care of their mother or father over the weekend or a week. [. . .] They technically weren't allowed to visit, but they were allowed to get in to provide care in crisis situations." (nurse, female, 54 years, primary level)

Successfully overcoming challenges through team creativity and deviation from established routines and structures was perceived as positive, even "inspiring" (health facility manager, male, 58 years, primary level).

## Discussion

This paper is the first to provide an in-depth analysis of the system-related factors that supported HCWs in Germany during the COVID-19 pandemic. Our analysis identified six categories, including: appreciation and encouragement; team support and communication; collaboration across professions and institutions; informational, material and technical support; leadership and participation; and creativity and innovation. By focusing on supportive factors, the analysis adds a salutogenetic perspective to the large body of evidence focused on stress, mental health and the negative consequences that the COVID-19 pandemic had on the health workforce and the health system [25–28]. Our analysis suggests that HCWs can demonstrate high self-efficacy and have the capacity to adapt in the face of a crisis, particularly when the environment and the system enables and supports such positive action. Through successfully handling the various challenges they faced during the COVID-19 pandemic, HCWs substantially contributed to the continued functioning of the German health system. More

importantly, however, our analysis highlights how crucial systemic and organisational factors, including organisational culture, conducive working conditions, supportive interactions between employers, employees and colleagues, and opportunities for participation, innovation and creativity are for health system resilience. These findings are in line with Abimbola and Topp who caution against the tendency to frame resilience solely at the individual level, which encourages individuals to "draw on internal strengths and resources to make up for weaknesses in the health system in the face of acute shocks or chronic stress" [29 p2], as it distracts attention and efforts away from addressing such systemic weaknesses. While we acknowledge that HCWs are not a homogenous cohort and that different types of HCWs experienced COVID-19 in different ways [30], our findings show that supportive factors at the organisational and systemic level are not specific to certain health facility types, professions or workplaces. Rather, organisational support was experienced by HCWs of all professions working in all types of health facilities, with no distinct patterns of more supported HCWs or more supportive workplaces emerging from the analysis. Our analysis, in fact, provides evidence that systemic and organisational support can be provided in any health facility and support any type of HCW.

Our analysis shows that the COVID-19 pandemic–in addition to posing huge challenges–offered extensive opportunities for innovation, creativity, and organisational change and transformation, and thus for health system strengthening. Ostensibly negative developments, like the malfunctioning and breakdown of previous hierarchical structures, enabled those working within the system to break from rigid routines and structures, try out new processes and develop creative and alternative solutions. Due to its cross-sectional nature, the study presented in this paper cannot draw conclusions about the ability of the German health system or the organisations within it to transform pandemic adaptations into long-lasting systemic change or about the sustainability of the changes that were made. It only draws attention to the potential for such transformation and offers a glimpse of what such transformation could entail. Yet, the findings support previous research which highlights that plasticity and transformation are key for resilience [31]. Our analysis describes multiple instances of innovation, adaptation and change during the COVID-19 pandemic, thus testifying to the German health system's resilience.

Barasa et al. conceptualise health systems as complex adaptive systems, and define resilience as "an active process within a dynamic health system that is constantly navigating challenges" [32 p4]. They argue that, in order to facilitate this process, conditions have to be created that enable "bottom-up dynamics" [32 p3] which allow development and innovation. Our analysis supports the concept of the health system as a complex system that adapts and learns, and the importance of bottom-up dynamics and self-initiative of those working within the system for the release of creativity, innovation and transformation. It further identifies several opportunities for learning from the adaptations that were made during the COVID-19 pandemic, which–if seized—are not only likely to contribute to health system resilience but also to health system transformation, organisational change, and the enhancement of HCWs' working conditions. Such adaptions would likely improve both the system's as well as individual HCWs' capacities to deal with future health system shocks. The implementation of such changes is likely to require the consideration and reform of a range of health systems functions, including governance, financing and human resources. It can be assumed that systemic support is needed in order to maintain any innovations that were initiated during the COVID-19 pandemic. A repeat study assessing current support and the maintenance of the changes described in our study (e.g. with regard to participatory leadership and organisational culture, interactions between employers, employees and colleagues, or working conditions) could investigate whether the necessary support for health system transformation is being provided and adaptations could be transformed into sustainable change.

We identify self-determined action and participation as key areas for health system transformation and learning. The findings presented above suggest that self-determination at work can increase motivation and facilitate creativity and innovation. According to Hackman and Oldman, a job with a high level of self-determination relates to the degree of independence that an individual has within their job and with regard to scheduling the work and determining work procedures [33]. In the context of the pandemic, HCWs positively stressed and valued opportunities for exerting self-determined action, e.g. in relation to decisions about implementing pandemic policies, ensuring the maintenance of health services and collaborating with colleagues from other disciplines. Previous studies on HCWs show that self-determination on the job is positively associated with psychological well-being [34], job satisfaction [35], and willingness and ability to stay in the job [36], whereas a lack of self-determination leads to reduced well-being [37], and thoughts of quitting the job [38]. Conducive work conditions are further assumed to increase self-determination at work and self-leadership [34] among HCWs. In line with this literature, our study identifies the participation of HCWs as crucial. Our analysis further suggests that establishing formats that allow interaction and exchange between those making managerial and regulatory decisions and those providing patient care is likely to increase not only satisfaction, engagement and self-determined acting at work, but also the adoption and implementation of policies and work processes that are actionable and in line with health care standards, and health system resilience. Participatory decision-making also promises to facilitate innovation, not only in pandemic times but also in routine care.

Participation and self-determination can only be achieved if the organisational culture values eye-level interaction and self-initiative [39]. Organisational culture is strongly influenced by leadership and management, so it is unsurprising that leadership styles and managerial support were identified as key supportive factors. Positively perceived assistance ranged from appreciation to practical support in the form of information, resources and technical training and to opportunities for communication, feedback and exchange of views. A supportive organisational culture further included interactions and exchange with team members and colleagues from other professions and institutions. The breaking down of boundaries between professions and institutions was perceived as facilitating collegial and effective working procedures, which helped to master pandemic challenges. There findings are in line with a recent OECD publication which highlights greater interdisciplinary task sharing and team work as a key area to increase health system resilience [40].

This study has number of limitations. Study participants' accounts of support were not validated via a quantitative questionnaire, which might have allowed a more objective measurement and the comparison of different types of support. Instead, the study provides in-depth insights into the factors that were perceived as supportive. Simultaneously, the cross-sectional study design did not allow the exploration of support over time or the sustainability of change and support. Future longitudinal studies could allow the analysis of support at different pandemic stages or a comparison of support provided to HCWs during the pandemic with support in non-pandemic times. Finally, the sample was restricted to key informants identified via media and other reports and HCWs volunteering to participate in the study. While accounts are likely to be influenced by self-selection bias, we are confident that the data was saturated and we captured key themes through the rigorous collection and analysis of the data. Purposive sampling further ensured that respondents were included from different geographical regions, working in various health facilities and other organisations, and holding a broad range of points of views, which makes it likely that the accounts represent a breadth of realities.

The analysis presented in this paper aligns with the existing literature which highlights the benefits that can be reached by increasing participation and self-determination, changing

organisational cultures, and improving the work conditions in the health sector [41, 42]. Our main finding is that systemic changes which enhance participation allow HCWs shape their work environment as well as work processes, and increase respectful and appreciative interaction between employers and employees and between colleagues of different professional groups have the potential to increase health system resilience. Implementing such changes might seem challenging in the hospital sector where hierarchies tend to be steep and rigid and procedures inflexible [43, 44]. However, evidence exists of health facilities successfully adopting alternative, less hierarchical work concepts and styles [41, 42]. This study suggests that the adaptation of such innovative ways of working by other health facilities could increase health system resilience.

## Supporting information

**S1 Table. Consolidated criteria for reporting qualitative studies (COREQ): 32-item checklist for the paper "The importance of organisational culture for health system resilience: A qualitative analysis of factors that supported health care workers in Germany during the COVID-19 pandemic".**
(DOCX)

## Acknowledgments

We thank Smilla Johann, Annika Witzigmann, Nadia Mangold, and Swetlana Muminow for their support in making this study possible while working as student assistants and interns in our research unit. We thank Angela Schuster, Sophie Müller, and Sameh Al-Awlaqi for study coordination and invaluable logistical support. We thank all participants for taking part in the study.

## Author Contributions

**Conceptualization:** Heide Weishaar, Charbel El Bcheraoui.

**Data curation:** Heide Weishaar, Megan Evans, Souaad Chemali, Brogan Geurts.

**Formal analysis:** Heide Weishaar, Megan Evans, Eloisa Montt Maray, Rike Böttcher, René Umlauf, Susan Abunijela, Nadine Muller, Barbara Buchberger, Hanna-Tina Fischer.

**Funding acquisition:** Charbel El Bcheraoui.

**Investigation:** Heide Weishaar, Megan Evans, Eloisa Montt Maray, Rike Böttcher, René Umlauf, Nadine Muller, Barbara Buchberger.

**Methodology:** Heide Weishaar, Megan Evans, Brogan Geurts, Hanna-Tina Fischer, Charbel El Bcheraoui.

**Project administration:** Souaad Chemali.

**Resources:** Charbel El Bcheraoui.

**Supervision:** Charbel El Bcheraoui.

**Validation:** Heide Weishaar.

**Writing – original draft:** Heide Weishaar.

**Writing – review & editing:** Heide Weishaar, Megan Evans, Souaad Chemali, Eloisa Montt Maray, Rike Böttcher, René Umlauf, Susan Abunijela, Nadine Muller, Barbara Buchberger, Brogan Geurts, Hanna-Tina Fischer, Charbel El Bcheraoui.

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
