## [Decision Letter · Decision Letter 0]

11 Jul 2024

PONE-D-24-20460The importance of organisational culture for health system resilience: A qualitative analysis of factors that supported health care workers in Germany during the COVID-19 pandemicPLOS ONE

Dear Dr. Weishaar,

Thank you for submitting your manuscript to PLOS ONE. After careful consideration, we feel that it has merit but does not fully meet PLOS ONE’s publication criteria as it currently stands. Therefore, we invite you to submit a revised version of the manuscript that addresses the points raised during the review process.

We look forward to receiving your revised manuscript.

Kind regards,

Prof. Anat Gesser-Edelsburg, Ph.D.

Academic Editor

PLOS ONE

Journal Requirements:

3. In the online submission form, you indicated that Data cannot be shared publicly because of confidentiality agreements with the participants and the data protection rules of the Robert Koch Institute. The data underlying the results presented in the study are available upon reasonable request from the corresponding author and would only be provided in a form that respects the confidentiality request of the participants and data protection rules of the RKI.

Reviewers' comments:

Reviewer's Responses to Questions

**Comments to the Author**

1. Is the manuscript technically sound, and do the data support the conclusions?

Reviewer #1: Yes

Reviewer #2: Yes

2. Has the statistical analysis been performed appropriately and rigorously? 

Reviewer #1: N/A

Reviewer #2: N/A

3. Have the authors made all data underlying the findings in their manuscript fully available?

Reviewer #1: Yes

Reviewer #2: No

4. Is the manuscript presented in an intelligible fashion and written in standard English?

Reviewer #1: Yes

Reviewer #2: Yes

5. Review Comments to the Author

**Reviewer #1:** The paper is well written and on a important topic, The paper follows a robust methodology and adhere to the guidelines regarding research ethics mentioned by the journal. I have suggested minor revision with comments attached separately

**Reviewer #2: **• Why were both in-depth interviews and FGDs conducted? How did they both serve the study objectives?

• How was the thematic analysis done? For example, the steps of coding and determination of themes.

• Clearly describe the kinds of Healthcare Workers (HCWs).

• Please outline a map of participant selection across Germany. For example, the number of primary, secondary, and tertiary healthcare facilities and where they are situated on the map of Germany. You may provide a simple table showing the number of participants selected from different healthcare facilities.

• A section of inclusion-exclusion criteria will add value to the paper.

• Maintain uniformity in quotes, especially who said it, for example, (Profession, gender, age, facility). Quotes are written in a separate paragraph and in italic style. However, the authors violate it on page no. 11.

• Authors should include the COREQ checklist (Tong et al., 2007) to enhance the study's reliability.

Tong, A., Sainsbury, P., & Craig, J. (2007). Consolidated criteria for reporting qualitative research (COREQ): A 32-item checklist for interviews and focus groups. International Journal for Quality in Health Care, 19(6), 349–357. https://doi.org/10.1093/intqhc/mzm042

6. PLOS authors have the option to publish the peer review history of their article (what does this mean?). If published, this will include your full peer review and any attached files.

Reviewer #1: No

Reviewer #2: **Yes: **Imteyaz Ahmad

---

## [Decision Letter · Decision Letter 1]

26 Dec 2024

The importance of organisational culture for health system resilience: A qualitative analysis of factors that supported health care workers in Germany during the COVID-19 pandemic

PONE-D-24-20460R1

Dear Dr. Weishaar,

We’re pleased to inform you that your manuscript has been judged scientifically suitable for publication and will be formally accepted for publication once it meets all outstanding technical requirements.

Kind regards,

Prof. Anat Gesser-Edelsburg, Ph.D.

Academic Editor

PLOS ONE

Additional Editor Comments (optional):

Reviewers' comments:

Reviewer's Responses to Questions

**Comments to the Author**

1. If the authors have adequately addressed your comments raised in a previous round of review and you feel that this manuscript is now acceptable for publication, you may indicate that here to bypass the “Comments to the Author” section, enter your conflict of interest statement in the “Confidential to Editor” section, and submit your "Accept" recommendation.

Reviewer #1: All comments have been addressed

Reviewer #2: (No Response)

2. Is the manuscript technically sound, and do the data support the conclusions?

Reviewer #1: Yes

Reviewer #2: Yes

3. Has the statistical analysis been performed appropriately and rigorously? 

Reviewer #1: N/A

Reviewer #2: N/A

4. Have the authors made all data underlying the findings in their manuscript fully available?

Reviewer #1: Yes

Reviewer #2: No

5. Is the manuscript presented in an intelligible fashion and written in standard English?

Reviewer #1: Yes

Reviewer #2: Yes

6. Review Comments to the Author

Reviewer #1: The authors have addressed my queries and made relevent changes to the manuscript, i recommend it for publication

Reviewer #2: The manuscript is well revised. It is good enough to be accepted for publication. Satisfactory literature review has been performed. Objective were clearly stated. Appropriate methodology has been used. Results were presented in a legible fashion. Discussion section provided arguments that supported the research findings. Limitations were also acknowledged. Policy recommendations were as per the findings of the study.

7. PLOS authors have the option to publish the peer review history of their article (what does this mean?). If published, this will include your full peer review and any attached files.

Reviewer #1: No

Reviewer #2: No

---

## [Editor Report · Acceptance letter]

3 Jan 2025

PONE-D-24-20460R1 

PLOS ONE

Dear Dr. Weishaar, 

I'm pleased to inform you that your manuscript has been deemed suitable for publication in PLOS ONE. Congratulations! Your manuscript is now being handed over to our production team.

Kind regards, 

on behalf of

Prof. Anat Gesser-Edelsburg 

Academic Editor

PLOS ONE